# A New Study on Features Exploring of the Concept of Wen and Zhi in Lao-Zhuang's Philosophy

**Shangkun Ji and Yongfeng Huang ***

Department of Philosophy, Xiamen University, Xiamen 361005, China; 10420200156438@stu.xmu.edu.cn
* Correspondence: yongfeng_huang1976@163.com

**Abstract:** For a long time, scholars have been applying the view of Confucius on Wen (文) and Zhi (質), centering on liyi (rites and rituals 禮儀) and renyi (benevolence and righteousness 仁義), to the Daoist concepts of them. This inevitably leads to many misunderstandings and overlooking the unique characteristics of the Daoist view. In the philosophy of Laozi and Zhuangzi, Zhi represents the natural state of human nature that is simple and desireless, while Wen refers to the corresponding expressions of speech and behavior generated based on Zhi. Under the Daoist Dao (the way 道)-Wu (object 物) model, the relationship between Wen and Zhi in Laozi and Zhuangzi is closely related to human nature and emotions, presenting Ti (source 體) and Yong (function 用) as the unity with nature as Ti and emotions as Yong. Secondly, Laozi and Zhuangzi's view on Wen and Zhi is closely related to the thoughts of self-cultivation and governing the country, with the latter as the foundation for the former. Their view of Wen and Zhi shows the relationship of Ben (root 本)-Mo (branch 末). The probing into the Laozi and Zhuangzi's concept of Wen and Zhi helps to understand the unique characteristics of the Daoist view, thereby further excavating the theoretical value and practical significance of the relationship between Wen and Zhi.

**Keywords:** Lao-Zhuang; View of Wen and Zhi (wenzhiguan 文質觀); Dao–Wu; source-function; root-branch



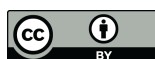

## 1. Introduction

The exploration of the view of *Wen* and Zhi is a crucial topic in the history of ancient Chinese philosophy, involving many aspects such as ways of living, social development, cultural qualities, and aesthetic values. *Wen* refers to the cultural aspect of humanity, including external cultural forms such as knowledge, education, etiquette, outer behavior, and literary embellishment; Zhi refers to the natural aspect of humanity, including inner virtues like essence, temperament, character, quality, substance, and basic stuff. The interaction between *Wen* and Zhi was considered by ancient Chinese thinkers as an important aspect of exploring the ideal way of living for human beings.[1]

The exploration of the relationship between *Wen* and Zhi began around the end of the Spring and Autumn period (770–476BC), and prospered during the mid-Warring States period (468–221BC). Even in the classics such as the *Book of Poetry* (*Shijing* 詩經), *Book of Documents* (*Shangshu* 尚書), and *Book of Changes* (*Zhouyi* 周易), there are already records related to the natural human state, cultural education, and other content of *Wen* and Zhi.[2] The *Discourses of the States, Jin 4* (*Guoyu* 國語·晉語四) records a conversation between Jin Wengong 晉文公 (about 697—628 BC) and his minister about education. The minister believed that "if someone is fundamentally good and guided by a virtuous mentor, then the cultivation of excellent talent can be expected. However, if one's nature is poor, even wise words of gold and jade cannot penetrate their heart, what good can their education do?" (Zuo 2002, p. 360). This reflects Zhi, referring to the nature of people and indicates that only by adopting corresponding educational methods based on a person's natural disposition, making *Wen* conform to Zhi, can the ideal results be achieved. The *Discourses of*

*the States-Jin5* (*Guanyu* 國語·晉語五) also records a person surnamed Ning Ying 寧贏 explanation of the relationship between appearance and language, which is "appearance is the glamour of a person's thoughts and feelings; speech is the elegance of the body. When language, imbued with grace, is expressed, actions should correspond with thoughts and feelings. A lack of cohesion among thoughts, language, and appearance indicates imperfection" (Zuo 2002, p. 376). Therefore, some scholars believe that linking emotion (*qing* 情), language (*yan* 言) and appearance (*mao* 貌) together properly reflects the continuous developmental pattern of the traditional Chinese relationship between *Wen* and Zhi (Hua 2020, p. 54).

During the pre-Qin period (770–221 BC), Confucianism began to focus on *Wen* and *Zhi*. Confucius 孔子 (551–479 BC) first put forward the concept of the a proper balance of *Wen* and *Zhi* (*wen zhi binbin* 文質彬彬). He said, "If basic stuff (zhi 質) exceeds literary embellishment (wen 文), one might be seen as uncouth; if refinement outweighs simplicity, one could appear superficial. Only when basic stuff and literary embellishment are properly balanced can one be considered as a gentleman" (*Analects* 論語 6.18) (Chen and Xu 2015, p. 68). He regards the Virtue–Rites (*de* 德-*li* 禮) relationship as the basis, believing that only when there is a correspondence between one's inner virtue and outer behavior can one embody the ideal moral character of a gentleman. After being reinterpreted by Confucius, *Wen* has come closer to the meaning of rites and ceremony (*liyi* 禮儀), and Zhi has been extended from human nature (*ziran benxing* 自然本性) to virtue of benevolence and righteousness (*renyi* 仁義). In *the Analects·wei ling gong*, Confucius once answered Zi Lu's 子路 (542–480 BC) question about how to be a mature person, "if one is wise like Zang Wuzhong 臧武仲, modest and restrained as Meng Gongchuo 孟公綽, brave like Bian Zhuangzi 卞莊子, and talented as Ran Qiu 冉求, further adorned with the virtues of rites and music (*yue* 樂), then one could be regarded as a well-rounded individual." (*Analects* 論語 15.12) (Chen and Xu 2015, p. 167). Wisdom, integrity, bravery, and talent are all manifestations of inner benevolence (*ren* 仁) and righteousness (*yi* 義), in addition to which there needs to be matching external adornment of rites and music. Therefore, in *Analects XianWen*, Confucius also said, "The gentleman values and emphasizes righteousness as essence, and practices it through the right conduct of ritual propriety (*liyi* 禮儀)." (*Analects* 論語 15.18) (Chen and Xu 2015, p. 189). This line defined Zhi as righteousness (*yi* 義), which is in line with rites (*li* 禮).[3] Confucius not only juxtaposed *Wen* and Zhi but also put them at the ends of each other's restraint, which interacted to regulate emotions. As it was said, "the practice of propriety was essential. Without it, one could not act."[4] (*Analects* 論語 1.12) (Chen and Xu 2015, p. 12).

In *Xunzi Lilun* (荀子·禮論), Xunzi 荀子 (313–238 BC) advocates the view that Zhi and Wen are interdependent, and proposes the concept that "Where rituals are elaborate but the emotions they express are simple, the ceremony is deemed grand; where rituals are simple, but the emotions they express are complex, the ceremony is seen as modest. Rituals and emotions should complement and blend with each other; this is the proper balance of ceremony. Therefore, a gentleman should conduct a grand ceremony when it should be grand, and a simple ceremony when it should be simple, and if a ceremony should be between grand and simple, then it should be conducted in moderation." (X. Wang 2013, pp. 422–23). In *Xunzi Wealthy Nation* (荀子·富國), Xunzi emphasizes that both Wen and Zhi should be appropriately balanced to regulate emotions. Xunzi sees the utilitarian function of adornment as a rational method for rulers to implement their governance. Thus, he argues: "For those who rule, without beautifying measures, they cannot unite the people. Without great rewards, they cannot manage their subordinates. Without authority or strength, they cannot prevent violence or subdue the unruly. Therefore, it is necessary to strike large bells, beat drums, blow ancient wind instrument, and play string to satisfy one's auditory enjoyment; it is necessary to engrave patterns on objects and draw designs on clothes to satisfy one's visual enjoyment; it is necessary to enjoy the meat of animals like cows, sheep, pigs, and dogs, fine grains, and various fragrant delicacies to satisfy one's gustatory enjoyment." (X. Wang 2013, pp. 212–13). In this way, the system of rites and music



promoted by the sage is seen as the best way to educate people in accordance with human nature.

Confucius initially established the concept of Wen and Zhi in the pre-Qin period, which was later enriched and expanded upon by Xunzi.[5] This perspective emphasizes the balanced utilization and mutual restraint of Wen and Zhi. Over time, scholars have continuously elucidated the essence of Wen and Zhi in Confucianism, leading to a deeper understanding of its meaning. It has found widespread application in various domains such as literature, art, and politics, eventually becoming the mainstream ideology regarding Wen and Zhi. The current academic discussion on the relationship between both is mainly based on the Confucian view of them; that is, under the De (virtue 德)–Li (rites 禮) model, the relationship between the inherent nature and the external form of people, events, and objects is discussed, in the hope of achieving the proper balance of Wen and Zhi.[6]

The Daoists of the pre-Qin period also had unique views on the relationship between Wen and Zhi.[7] However, for a long time, influenced by the mainstream Confucian view, the Daoist view of them has been discussed under the research framework of the Confucian view. This has resulted in many shortcomings and misunderstandings in the research and application of the Daoist view. The most prominent manifestation is that the Daoist focuses more on Zhi based on the Dao–Wu model, which is understood as the opposition and antagonism between Wen and Zhi under the research framework of the Confucian view. For example, Shu Jianhua's 舒建華 (Shu 1992, pp. 45–50) article, the topic of which is "*Symbol of Wen and Zhi: An Interpretation of Culture Studies—An Overview of Confucian, Daoist, and Mohist Views on Wen and Zhi*," early observed the characteristics of the Daoist view of Wen and Zhi, distinguishing it from the views of Confucianism and Mohism. He summarized the Daoist emphasis on the ontology of the universe and the importance of nature over embellishment as "rejecting Wen in favor of Zhi" (*yi zhi fan wen* 以質反文), aimed at opposing all artificial decoration and achieving the highest aesthetic realm. Li Xiongyan's 李雄燕 (Li and Li 2012, pp. 24–27) article, titled "*Favoring Zhi, Diminishing Wen: Analysis of the Perception of Zhi and Wen in the 'Nanhuazhenjing' Anno-tations Group from the Perspective of 'Others'*," examines the promotion of Zhuangzi's aesthetic claim of "favoring Zhi, diminishing Wen" (*shang zhi qing wen* 尚質轻文) by Daoist scholars over generations. In "*A Brief Account of the Aesthetic Thought of the Heavy Metaphysics School in the Tang Dynasty: A Case Study of Cheng Xuan Ying and Li Rong*" by Li Pei 李裴 (F. Li 2018, pp. 62–66), it is proposed that the Daoist scholars of the Tang Dynasty, represented by Cheng Xuanying 成玄英 and Li Rong 李榮 who emphasized the beauty-truth (*mei–zhen* 美-真) relationship, enriched and supplemented the Confucian view of goodness–beauty (*shan–mei* 善-美). They showed an overall attitude of valuing Zhi over Wen. Tian Chun's 田春 (Tian 2010, pp. 50–52) "*Discussion on the Relationship between Usage and Decoration in the View of Wen and Zhi of Pre-Qin*" suggests that Daoism does not entirely oppose the Confucian restricted view of the relationship between Wen and Zhi, or the view that seeks to balance aesthetics and functionality. Instead, Daoism advocates following the principles of "subordinating Wen to Zhi" (*yi zhi yue wen* 以質约文) and "the Wen conforming to the Zhi" (*wen shunying zhi* 文顺应質). Lai Xisan's 賴錫三 (Lai 2019, pp. 1–24) article, "*Dialects between Quality and Form and the Ethical Revaluation---Zhuangzi's Critical Reflection on True Meaning of Rites*," explores a novel perspective on the interpretation of "rites" in the Zhuangzi text. Lai posits that Zhuangzi's criticism of "rites" should not be understood as an outright opposition to ritual education. Rather, Zhuangzi aims to rejuvenate and adapt the rigid ritual system by reflecting on the concept of "rites" through the lens of form and quality dialectics.

There are several fundamental problems with the current research on the Daoist view of Wen and Zhi. Firstly, there has been no detailed clarification of the Daoist concept of Wen and Zhi, with most discussions being based on the Confucian interpretation of these terms. Secondly, much of the research tends to discuss them on the basis of Confucius's model of virtue and rites, while turning a blind eye to considering the Daoist model of Dao–Wu. Thirdly, when discussing the Daoist relationship between Wen and Zhi based

on the Confucian principle of balance and moderation, the Daoist preference for Zhi is often interpreted as an oppositional and mutually exclusive relationship of Wen and Zhi. This leads to a misreading of the Daoist view as the antagonism between both and results in the loss of its distinctiveness. All this shows that the current academic research on the Daoist view is still unable to break free from the framework of related Confucian studies, which inevitably restricts the exploration and understanding of the true value of the Daoist view of Wen and Zhi.

To solve the problems mentioned above, this article intends to explore the characteristics of early Daoist thoughts on Wen and Zhi as represented by Laozi and Zhuangzi, from three aspects. Firstly, by sorting out the concepts of Wen and Zhi in Laozi and Zhuangzi and redefining their connotations and extensions. Secondly, based on a clear understanding of the concepts of Wen and Zhi in Laozi and Zhuangzi, an interpretation is provided for the embodied and utilitarian relationship of Wen and Zhi as presented in the Dao–Wu elucidating its inherent characteristic of source and funciton as the unity (*ti yong yiyuan* 體用一元). Thirdly, the research focuses on interpreting the characteristics of Ben (root)–Mo (branch) as one (*ben mo yiti* 本末一體). Through these three analyses, the unique concept of Wen and Zhi in Laozi and Zhuangzi is summarized. Additionally, by analyzing and arguing in the aforementioned three areas above, we transcend the ideological framework of the Confucian concept of Wen and Zhi and summarize the distinctive concept of Wen and Zhi in Laozi and Zhuangzi.

## 2. The Concept of Wen and Zhi in Lao-Zhuang Philosophy

Lovejoy believes that the elements of philosophical doctrines, in differing logical combinations, are not always readily recognizable. Therefore, philosophical semantics can be applied to discern the differences and functions among these concepts (Lovejoy 1933, pp. 4, 14). On the surface, it seems both Confucius and Lao–Zhuang are discussing, through their perspectives on Wen and Zhi, the relationship between a person's inner essence and external form. However, due to their different fundamental points, their specific scope and methods determine the essential differences in the connotations of their Wen and Zhi. Therefore, we need to redefine Laozi and Zhuangzi's concepts of Wen and Zhi.

### 2.1. Wen in Lao–Zhuang Philosophy

The oracle bone script 夨 for Wen implies the meaning of decorated body (*wenshen* 文身). In *Shuowenjiezi* 說文解字 (Z. Xu 2006, p. 996), "it looks like a standing human who is figured with patterns which was carved on the chest, hence the patterns on the decorated body will be seen as characters." Gao Wenqiang 高文強 (Gao 2021, p. 4) believes that Wen reflects the root meanings at three levels. First, the earliest Wen refers not only to what is man-made but also to patterns drawn on the human body. It means human and Wen are inseparable. Second, Wen is the earliest "carved decoration" (*kehua de wenshi* 刻畫的文飾) and the cultural and artistic text created by humans. Third, Wen is perceivable, observable, sensible, and knowable, so it is the object of human perception. It can be seen from this that Wen refers to patterns and designs inscribed on tangible objects and is also a representation of one's self-identity. This implies that Wen and material objects should be understood as a unified whole in terms of meaning.

Laozi expounded the concept of Wen. In the *Daodejing* Chapter 19, he noted that "Banish sagehood, discard wisdom, (jueshengqizhi 絕聖棄智)and the people will benefit a hundredfold. Banish benevolence, discard righteousness (juerenqiyi 絕仁棄義), and the people will become dutiful and compassionate. Banish cunning, discard profit (jueqiaoqili 絕巧棄利), and thieves and robbers will disappear." These three aspects, sagehood and wisdom (*shengzhi* 聖智), benevolence and righteousness (*renyi* 仁義), and cunning and profit (*qiaoli* 巧利) are all superficial Wen. As a code for addressing societal issues, they are not sufficient (*cisanzhe weiwen buzu* 此三者為文不足)". (*Daodejing Chapter 19*) (G. Chen 2015, p. 128) and to discard the ostentatious wisdom and rites in order to be free from worries.

He categorizes sagehood and wisdom, benevolence and righteousness, and cunning and profit as Wen, which should be completely banished.[8]

Moreover, sagacious wisdom refers to the wisdom of the sages. In Laozi's view, the way the sages educate the people should be to act without contriving and to teach without words (*Daodejing* Chapter 2) (G. Chen 2015, p. 58), rather than imposing norms, rites, and other contrived methods. "The true sage is the one who possesses Dao and never seeks great contributions, which is why they can accomplish great deeds." (*Daodejing* Chapter 63) (G. Chen 2015, p. 282). Laozi describes the sage as "dressed in coarse cloth but holding a gem in his heart." (*Daodejing* Chapter 70) (G. Chen 2015, p. 303). They do not pursue outward decoration but uphold the simplicity and tranquility of the inner spirit. Wisdom and benevolence are not worth promoting in Laozi's view. His words, "when the great Dao is banished, benevolence and wisdom appear" (*Daodejing* Chapter 18) (G. Chen 2015, p. 126) indicate that using these two to educate the people and govern the world not only fails to achieve the intended goals but can also be detrimental to the people and the state. This is as he says, "The difficulty in governing the people is due to their excess of cunning. Therefore, using cunning to govern a state is being a bandit of the state; not using cunning to govern a state is being a blessing of the state." (*Daodejing* Chapter 65) (G. Chen 2015, p. 288). Cunning for profit is a major cause of restlessness in people's minds and chaos in the state. Laozi says, "When people have many clever tools, strange things start to appear; when laws and regulations are excessively detailed, more thieves and bandits appear." (*Daodejing* Chapter 57) (G. Chen 2015, p. 264). Cunning for profit increases people's desire for external things, while the complex laws and norms that serve governance only make society more confused. It can be seen that sage wisdom, benevolence and righteousness, and "cunning for profit" are all Pseudo-Wen and are not proper for governing and educating the people.

What Laozi objects to is Pseudo-Wen instead of the true one. One proof of this is that in the sentence "Wei Wen (為文) is not sufficient to educate people," Wei (為, the second tone in Chinese pinyin) is similar to another Chinese character Wei (偽, the third tone in pinyin), meaning the false, in terms of pronunciation and form. Another evidence is that in the book of Laozi written by Guo Dianchujian 郭店楚簡, which appeared earlier than the existing version, " Banish benevolence，discard righteousness" (絕仁棄義) was changed into " Banish fraud, discard false" (juezhaqiwei 絕詐棄偽).[9] In Laozi's thought, the real purpose of educating people with Wen is not to manage and regulate the people, but to make people desireless and unambitious, in harmony with natural law and morality. Laozi says: "Therefore, it is necessary for people's thoughts and perceptions to have an anchor-to maintain a pure and simple nature, to reduce selfish desires and distractions, and to discard the ostentatious wisdom and rites in order to be free from worries (故令之有所屬: 見素抱樸, 少思寡欲)."[10] (*Daodejing* Chapter 19) (G. Chen 2015, p. 128). This calms the restless and greedy hearts of the people, returning them to the simplicity and tranquility of the inner spirit. This indicates that beyond the deceitful Pseudo-Wen, there exists a Wen that is simple and free of self-interest. This True-Wen stands in contrast to the Pseudo-Wen. This can explain why Laozi opposes rites, considering them a product of insufficient loyalty and faith, and the beginning of chaos and disorder. Yet, he also said, "After a great battle, mourning rites should be used to deal with it." (*Daodejing* Chapter 31) (G. Chen 2015, p. 177). Different from the rites spoken of by Confucius, the rites that Laozi opposes refer to those separated from righteousness, that is, they only refer to external rituals or some constraints, which belong to the Pseudo-Wen, and therefore need to be discarded. Laozi does not oppose the rites that incorporate Dao (the way 道), De (virtue 德), Ren (benevolence 仁), and Yi (righteousness 義). These rites can be understood as True-Wen, which should be followed when necessary. In fact, Later commentators also noticed Laozi's opposition to *Pseudo-Wen*. For example, from the perspective of the discrepancy between the integrity of value presuppositions and the limitations of human behavior and experience, Wang Bi 王弼 (226–249 AD) views " sagehood, wisdom, benevolence and righteousness" as value presuppositions that people cannot fully understand and perfectly achieve in re-

ality. If they are viewed as simple decorations, it would be a hypocritical act (Lynn 1999, p. 82).[11] In other words, what Laozi rejects is not sagehood, wisdom, love, and justice, but those with ornamentation. He believes people should remain simple, unadorned, selfless, and desireless (D'Ambrosio 2022, p. 8).

Zhuangzi's understanding of *Wen* carries on from Laozi, enriching its connotation and linking it to human emotions and dispositions. In *Zhuangzi Xiaoyaoyou* 莊子·逍遙遊, there is a description of the "The Yue cut their hair short and painted on their bodies (*yueren duanfa wenshen* 越人斷髮文身)" (G. Chen 2016, p. 29). Here, the "tattoo of the Yue people" means "Wen", and stands for ornamentation. In *Zhuangzi Renjianshi* 莊子·人間世, there is an allegory of a stonecutter dreaming of a tree. The tree asks the carpenter, "How do you compare me? Do you compare me to a Wen tree?" (G. Chen 2016, p. 139). Here, Wen implies human intervention and utility, and a *Wen* tree means a decorative and useful tree. He also said, "For a tree that has grown for hundreds of years, when it is cut down and opened, it can be made into a wine vessel for worship, with green and yellow patterns painted on it, and the remaining wood is thrown into the ravine. If you compare the tree made into a wine vessel with the wood thrown in the ravine, there is a difference in status between them, but they both lost their nature." (G. Chen 2016, p. 343). Zhuangzi believes that although the trees decorated by humans are beautiful, they have also lost their inherent nature and essence. In *Zhuangzi Yingdiwang* 莊子·應帝王, he contrasts *Wen* and *Shi* (essence 實) and says, "What I have taught you is *Wen* instead of *Shi*. How could you truly attain the *Dao*?" (G. Chen 2016, p. 227). This indicates that *Wen* represents the appearance of *Dao* instead of its essence. In addition, Zhuangzi further connects it with human nature and emotions.

In Zhuangzi's teachings, he compares artisans to saints, claiming that it is a sin for artisans to spoil raw materials to make tools, and it is a mistake for saints to destroy Dao and De to impose benevolence and righteousness (G. Chen 2016, p. 258). In *Zhuangzi Mati* 莊子·馬蹄, he posits that the loss of people's inherent nature is the result of saints using decorative teachings. Zhuangzi states, "As for the saints, they stumble with benevolence and hobble with righteousness, and the world starts to be doubtful. When the tunes are spread far and wide for joy, and when judgments are made in all quarters about right and wrong, the world starts to be divided. As for who can make it so that there are no damaged materials made into sacrificial vessels? Who can make it so that jade is not broken up to make scepters? If Dao and De are not destroyed, why would we need benevolence and righteousness? If our natural inclinations and desires are not driven away, why would we need rites and music? If the five colors are not confused, who will make up the decorations? If the five tones are not confused, who will tune the six musical notes?" (G. Chen 2016, p. 258). He believes that benevolence, righteousness, rites, and music are all human-made Wen, which deviate from Dao and De. In *Zhuangzi Pianmu* 莊子·駢拇, he also says: "Toes growing together or an extra finger growing on a hand, to human nature, are akin to behaviors indulging and being obsessed with benevolence and righteousness, and like using human hearing and vision beyond their normal capacities. So those with particularly good vision who disrupt the five colors and confuse the patterns, aren't they just like the eye-catching patterns of green and yellow on ceremonial robes? Li Zhu 離朱 is a representative of these people." (G. Chen 2016, p. 242). He asserts that actions of benevolence and righteousness should be in accordance with human nature. Misusing cleverness to over-decorate benevolence and righteousness not only deviates from the natural Dao but also disrupts people's inclinations and desires. Such actions of benevolence and righteousness not only fail to educate people but also constrain their actions and harm their inherent nature. Zhuangzi identifies Wen that corresponds to human nature as the genuine Wen, which is the righteousness of Dao and De, and opposes the Wen that is over-decorated by misusing cleverness, which he calls Pseudo-Wen. This view is consistent with Laozi's. In *Zhuangzi Zaiyou* 莊子·在宥, Zhuangzi said,

> Are you fond of benevolence? This is disturbing to one's nature. Fond of righteousness? This is contrary to common sense. Fond of propriety? This promotes

cunning. Fond of music? This encourages indulgence. Fond of sainthood? This aids in craftmanship. Fond of wisdom? This fosters nitpicking. (G. Chen 2016, p. 284)

說仁邪，是亂於德也；說義邪，是悖於理也；說禮邪，是相於技也；說樂邪，是相於淫也；說聖邪，是相於藝也；說知邪，是相於疵也。

He opposes the Pseudo-Wen of benevolence, righteousness, rites, music, and knowledge (wisdom). However, he also mentions that the ancient true men used "punishments as the foundation, rites as the supplements, knowledge to seize the opportunity, and virtue as the guide." in *Zhuangzi Dazongshi* 莊子·大宗師 (G. Chen 2016, p. 177). What he advocates is the natural law, righteous conduct, and moral virtues that coincide with the essence of Dao, which all belong to True-Wen. These are all necessary for a true person to act in the world while remaining calm within.

From the above, we can see that Laozi and Zhuangzi discussed Wen based on the premise of human nature, and rejects Pseudo-Wen that contradicts natural nature and is mixed with personal desires, such as deceptive speech, gorgeous cultural decorations to satisfy personal desires, rites and music that have lost their moral righteousness. Correspondingly, Laozi endorses True-Wen that is in harmony with nature, in line with the original, unadorned, and stripped of external manifestations of personal desires.

### 2.2. Zhi in Lao–Zhuang Philosophy

The term Zhi 質 originally referred to the act of pawning something (Zhang 2001, p. 732). According to *Erya* (爾雅), an ancient Chinese dictionary, Zhi means trust (Chi 2008, p. 190). In *Shuowen Jiezi* (說文解字), it means to exchange with real objects, derived from the symbols for shell and metal. As Duan Yucai 段玉裁 commented: "Exchange with real objects is just like exchange hostages between vassal states in the Spring and Autumn Period in order to gain trust from each other. Later, it took on extended meanings of 'simplicity' (pu 樸) and 'earth' (di 地), such as in the phrase 'having substance, having function.'" (S. Xu 1981, p. 281). In other words, the original meaning of Zhi is to mortgage with real objects or people. *Shuowen Jiezi Commentary* (說文解字詁林) noted Zhi is actualities (*shi* 實). When doubts arise, people or things are used as a pawn to dispel suspicions (Ding 1988, p. 6502). Therefore, Zhi has connotations of being simple, honest, and unadorned. It is often contrasted with Wen and represents the original state of a thing before it has been processed or decorated. From this, we can understand that Zhi reflects two fundamental aspects: the inherent nature of things, which is the basis of decoration, and the earliest human interaction with the world, which is free from deceit.

Laozi pays great attention to Zhi. In *Daodejing* Chapter 41, he elaborates his understanding of Zhi by saying, "The way out into the light often looks dark. The way that goes ahead often looks as if it went back. The way that is least hilly often looks as if it went up and down, the "power" that is really loftiest looks like an abyss. What is sheerest white looks blurred. The "power" that is most sufficing looks inadequate. The "power" that stands firmest looks flimsy. What is in its natural, pure state looks faded." (G. Chen 2015, p. 212). He here connects and equates a series of opposing concepts, while the phrase "natural and pure state looks faded" (質真若渝) could help us understand what Zhi is and nature carries the meaning of simplicity. He uses simplicity to describe the natural state of human nature. He also said, "being honest and simple, just like a piece of uncarved material" (*Daodejing* Chapter 15) (G. Chen 2015, p. 111) and "use the true simplicity of Dao to pacify it (the emerging desire), and there will be no desire." Just as Laozi said "*baopu* 抱樸"[12] "When desire does not arise and a state of peace is achieved, and the world will naturally settle." (*Daodejing* Chapter 41) (G. Chen 2015, p. 194). He believes that simplicity can halt human desires, return human nature to tranquility, and the world will follow suit. He also says, "a man bearing virtue is enough to return to simplicity" (*Daodejing* Chapter 28) (G. Chen 2015, p. 165). He points out that people should preserve noble virtues, so they can return to the natural state of simple life. From this point of view,

simplicity refers to the state of human nature with no desires, no thoughts, no actions, nature, silence, etc. Zhi also has a meaning of truthfulness, which is closely related to the connotation of truth. In *Daodejing* Chapter 21 Laozi says: "Although it is tiny and hard to perceive, there is essence within it. The essence there is so real, and it can be confirmed." (G. Chen 2015, p. 138). He uses truth to describe Dao as a verifiable reality. Then "Zhi is true" means that human nature is also a perceptible and verifiable real state. "Yu" (渝) describes the state of clear pond water being stirred and becoming turbid. Because the phrase "Zhi is true as Yu" uses the interpretation method of contrasting opposites, it shows that the meanings of Zhi and Yu are opposite. Zhi can be understood as the antonym of Yu, that is, states such as tranquility, stillness, immutability, and clarity.

Based on Laozi, Zhuangzi's interpretation of Zhi is clearer. In his view, tranquility, loneliness, emptiness, and inaction are the laws that make heaven and earth balanced and change, and also the essence of Dao and De (G. Chen 2016, p. 406). He uses the characteristics of water to provide a more specific explanation of Zhi. He says: "The nature of water is such that it becomes clear when there are no impurities, and it remains calm when not stirred; but if the water encounters obstruction and cannot flow, it cannot become clear, which is one manifestation of water's inherent nature." (*Zhuangzi Keyi* 莊子·刻意) (G. Chen 2016, p. 409). This is exactly the same as what Laozi said above. Zhuangzi also believes that Zhi has the characteristic of simplicity. He says: "What is meant by plain is that it is not mixed with anything else; what is meant by pure is that it does not lack its spirit. Those who can embody purity and plainness are called true men." (G. Chen 2016, p. 410). This is to teach people to be indifferent to fame and fortune, without private desires and arbitrary actions, which is in line with natural laws and the way to nourish the spirit. This is exactly what Laozi called "plainness" (*pusu* 樸素). He also says: "When everyone refrains from cunning, our innate nature is not likely to be lost. When everyone refrains from greed, we all retain a sense of plainness and simplicity. Remaining pure and simple allows us to maintain our human nature. However, when sages appear, overly eager for benevolence and righteousness, confusion begins to spread across the world; when indulgence in pleasure and tedious ceremonies are pursued, the world begins to divide." (G. Chen 2016, p. 258). Zhuangzi also sees ignorance and no desire as human nature. He opposes the sages to use benevolence, righteousness, ritual, music, and other human-made etiquettes, norms, and superficial forms to restrict and disturb people's simple natural nature. Zhuangzi believes ritual education is an act of the sages and damage to human nature. From the above, it can be seen that Zhi mentioned by Zhuangzi is the same as the one mentioned by Laozi, and it is a state of mind of tranquility, solitude, emptiness, inaction, ignorance, and desirelessness, which comes from the nature of Dao and De and is the essential attribute of human spirit.

The above shows that the concepts of Wen and Zhi discussed by Laozi and Zhuangzi are different from the Wen referring to external ritual, and Zhi, referring to internal benevolence and righteousness emphasized by Confucius. If the concepts of Wen and Zhi in Laozi and Zhuangzi are not clearly and systematically sorted out and the Confucian concept of Wen and Zhi is used to interpret the connotation of Daoist ones in a one-sided way, it may not only lead to a deviation in Daoist understanding, but also ignore the characteristics of Daoism, and it will not be possible to correctly explore the relationship between the Daoist Wen and Zhi.

### 3. The Relationship of Source and Function in Lao–Zhuang's Concepts of Wen and Zhi

Starting from human nature as the logical starting point, the relationship between Wen and Zhi discussed by Laozi and Zhuangzi can be seen as Ti (source)-Yong (function) as the unity based on the Dao–Wu model.[13] (Ye 2017, p. 31) (Matthews 2021, p. 260). Their exploration of the relationship between culture and quality is actually to seek an ideal way of life in the world, which is the natural state of harmony between human nature and all things. Laozi believes that Dao connects all things, and the relationship between Wen and Zhi is based on this Dao–Wu model. In *Daodejing* Chapter 25, Laozi said, "There are things

mixed and created, born before heaven and earth" (G. Chen 2015, p. 151) and "The Dao gives birth to all things and events and the virtue nourishes them. As a result, a variety of forms emerge from all things and events, and everything grows. Therefore, all things and events universally respect Dao and cherish the De." (*Daodejing* Chapter 51) (G. Chen 2015, p. 243). He abstracts Dao as the ontology of all things. It is an absolute and independent existence and the first cause of all things.

Laozi believes that in the process of generating all things, Dao gives all things their inherent qualities. Therefore, Dao exists in all things and influences the development of all things as a natural law. Just as Wang Bi's commentary says, "Simplicity is true. When truth is scattered, hundreds of behaviors emerge, and different kinds of objects are born like tools." (B. Wang 1980, p. 75). Strictly speaking, however, concrete things do not embody all of the Dao. Laozi said, "It is only after losing Dao that De arises." He called the Dao given to concrete things virtue, referring to the nature of all things. After all things are generated, they continue to develop relying on their own nature. Virtue has thus become the inner basis for the survival of all things. It is worth noting that as the intermediate link in the development model of Dao–Wu, De can merge with Dao upwards to represent the universality of all things; it can merge with object downwards to represent the specificity of concrete things distinct from other things. For example, Laozi and Zhuangzi have different expressions to explain the uniqueness represented by virtue, such as upper virtue (*shangde* 上德), lower virtue (*xiade* 下德), constant virtue (*changde* 常德), heavenly virtue (*tiande* 天德), and so on. They believe that the return of virtue to Dao can make the inherent qualities of Dao in things perfect while the development of virtue towards objects will gradually damage the inherent qualities of Dao in things.

Laozi and Zhuangzi's exploration of the relationship between Wen and Zhi is often associated with the unique nature of human beings that distinguishes them from other things.[14] According to Laozi and Zhuangzi, De, the attribute of Dao embodied in humans, is sometimes also referred to as nature. Laozi said, "a man bearing virtue is enough to return to simplicity." (*Daodejing* Chapter 28). He believes that people should maintain a simple and unadorned nature in order to return to the natural original pure state. Zhuangzi said, "Dao is the master admired by virtue, life is the radiance released by virtue, and nature is the essence of life." (*Zhuangzi Gengsangchu* 莊子·庚桑楚) (G. Chen 2016, p. 623). This indicates that nature refers to the essential attributes of a person, which come from Dao and De. While interpreting Chapter 51 of *Daodejing*, Yan Zun 嚴遵 (about 82 BC–10 AD) said, "Dao gives form to all things, resulting in different types such as people of different genders and characters and objects of various sizes; these are referred to as nature. What is conferred by De includes wealth, nobility, poverty, lowliness, longevity, pain, pleasure, appropriateness and inappropriateness; these are referred to as destiny. ... acting according to nature, one engages with the external world and gains insights, love, hate, surprise, fear, happiness, anger, sorrow, joy, worry, resentment, ambition, retreat, misery, generosity; these are called emotions. ... one is entangled in different types of things, being concerned about various matters and being tied up together; these are called desires." (Z. Yan 2013, p. 90). He interprets nature as different human characteristics, making it more specific. Nature comes from Dao and De. If it is triggered by the outside, then emotions arise, which gives one no way to escape; thus, desires emerge. Zhuangzi believes that an ideal life state is the coexistence of nature and emotions. In *Zhuangzi*, He said, "The true Dao is righteous, which means not losing the inherent nature of human beings. Those who are not benevolent indulge in desires and greed for glory, wealth, and honor." (G. Chen 2016, pp. 245–46). This means that when nature and emotions are in harmony, their words and deeds are also in line with morality. If nature and emotions are apart, they will fall into desire. Zhuangzi also refers to the state of coexistence of both as the state of De. He said, "The ordinary has an unchanging nature, which is to weave cloth, then wear clothes and to cultivate crops, then have food. This is called De" (*Zhuangzi Shanxing* 莊子·繕性) (G. Chen 2016, p. 257).

These are all about the importance of nature and emotions pursuing human virtue. (Z. Chen 2012, p. 136) Zhuangzi also believes that desires will cause damage to nature. In

Zhuangzi, *Xuwugui* 徐無鬼 told Wei Wuhou 魏武侯, "If your desires inflate fueling feelings of likes and dislikes, then your emotions will be harmed." (G. Chen 2016, p. 631). Desire will cover up people's quiet and effortless nature, and the emotions that deviate from nature will become more and more chaotic, and then give birth to more desires. The relationship between Wen and Zhi explained by Laozi and Zhuangzi revolves around human nature. For example, the Pseudo-Wen opposed by Laozi and Zhuangzi refers to all kinds of hypocrisy, superficial etiquette, norms, and behaviors that are produced by the deviation between human nature and emotions. In contrast, the True-Wen advocated by Lao Zhuang refers to the way of speaking and acting that naturally arises from the unity of human nature and emotion. In the context of Laozi and Zhuangzi's relationship between Wen and Zhi, the former is positioned as usefulness and manifestation, naturally derived from the Zhi, while the latter is at the root position, referring to the state of nature being quiet and desireless. Both are inseparable parts of the whole. Then, how is the relationship between them shown in people's nature and emotions? Zhuangzi said,

> The movement of nature is called behavior (wei 為), and behavior resulting from human desire is called error(wei 偽). Knowledge (zhi 知), as it is understood, is the response to external things and is the planning (mou 謀). Seeking knowledge yet acknowledging the unknown, is like the limited view one gets when looking at things with a squinted eye. Actions driven by necessity are called "virtue" (de 德), and every action that does not deviate from my nature is called governance (zhi 治). Although virtue and governance are antonyms in name, their essence is consistent. (*Zhuangzi Gengsangchu* 莊子·庚桑楚) (G. Chen 2016, p. 623)

> 性之動謂之為，為之偽謂之失。知者，接也。知者，謨也。知者之所不知，猶睨也。動以不行已之謂德，動無非我之謂治，名相反而實相順也。

Zhuangzi believes that the reason for Wen arising from Zhi is the knowledge coming from nature. When the serene nature, akin to calm water, is stirred by external things, it creates ripples which symbolizes knowledge. Knowledge is the differentiation of things. If one adheres strictly to the distinction between things and between oneself and things, desire arises. And this is also what Yan Zun referred to as the desire that is "entangled with all things and yet difficult to dissolve." (Z. Yan 2013, p. 90). With desire, people will produce corresponding speech and behavior, that is, action. If these actions deviate from their innate nature, it will give rise to a variety of insincere and pretentious activities, leading to the loss of one's moral character. As one's knowledge increases, he will have more desires and anxieties, and then, the originally calm nature will become chaotic, drifting further away from Dao. This is what Laozi meant when he said, "Those who pursue learning increase their knowledge day by day, while those who pursue the great Dao reduce their desires day by day." (*Daodejing* Chapter 48) (G. Chen 2015, p. 233). There is a similar description in Zhuangzi,

> people do good by deviating from Dao and their virtue, and then they abandon their nature to succumb to desires. As the mutual understanding between hearts cannot stabilize the world, they add complicated rhetoric and enrich various kinds of knowledge. However, it should be known that rhetoric annihilates nature and knowledge drowns the soul (wenmiezhi, bonixin 文滅質, 博溺心). In this way, people begin to be confused and disturbed, thus unable to return to those who should be in nature. (*Zhuangzi Shanxing* 莊子·繕性) (G. Chen 2016, p. 415)

> 離道以善，險德以行，然後去性而從於心。心與心識知，而不足以定天下，
> 然後附之以文，益之以博。文滅質，博溺心，然後民始惑亂，無以反其性情而復其初。

According to Zhuangzi, once a person departs from nature and follows the heart, it implies that they continue to indulge in desires by clinging to external things, which leads to mutual suspicion and speculation among people. At this time, using Pseudo-*Wen*, such as etiquette and norms, to educate people will inevitably fail to achieve the desired effect

because using Pseudo-*Wen* that has departed from *Zhi* to restrain it will not only fail to achieve good results but will also cause damage to human nature. This is also what Laozi referred to as "To benefit brings harm" (*Daodejing* Chapter 42) (G. Chen 2015, p. 216), and what Zhuangzi meant by "Pseudo-*Wen* destroys *Zhi*, and knowledge drowns the soul..." thus, leaving people unable to return to those who they should be in nature.

However, people inevitably have to interact with people and things around them. In this case, how can one return to the pure state of original nature? Laozi suggests that one should abandon learned knowledge and eliminate insincere speech and behavior to finally reach the ideal state of no distinction between self and objects (G. Chen 2015, p. 131). Moreover, Zhuangzi suggests that one may nurture the mind with tranquillity, and harmonize tranquillity with the mind. The mind and tranquility mutually regulate each other; thus, a harmonious and compliant disposition is revealed from within one's nature (*Zhuangzi Shanxing* 莊子·繕性) (G. Chen 2016, p. 413). He also believes that we should work on the mind, strive to reduce the accumulation of worldly knowledge, and eliminate the distinction between self and objects as much as possible to respond to the disturbance of external things and reduce thoughts and desires. Doing so will make virtues such as benevolence, righteousness, propriety, and music naturally emerge from human nature. Jiang Xichang believes that Laozi's analysis of desirelessness has two meanings: one refers to the desirelessness in the era when all things were nameless, which means the complete absence of desire; the other indicates the one in human beings in the era when all things are named, which aligns with the notion of embracing minimalism and curbing desires, and it is not to eliminate all desires. He thinks that the state of absolute desirelessness represents the essence of Dao and De, while embracing minimalism and curbing desires refer to the fundamental nature of human beings (Jiang 1937, p. 9). This natural state of life is a reflection of human nature. It is the manifestation of True-Wen coming from Zhi. Zhuangzi believes that to achieve this ideal state of life, and one should keep in mind that,

> There's nothing better in behavior than to comply with nature, and there's nothing better in feelings than to be sincere and genuine. To comply with nature means not to contradict external matters; to be sincere and genuine means your spirit will not be exhausted. Without contradiction and exhaustion, there's no need to embellish yourself with Wen, and then there's no need to rely on external things. (*Zhuangzi Shanxing* 莊子·繕性) (G. Chen 2016, p. 521)
>
> 形莫若緣，情莫若率。緣則不離，率則不勞。不離不勞，則不求文以待形。不求文以待形，固不待物。

This means that we should follow objective rules in life, keep our hearts consistent with our words and deeds, and learn to be content with what comes our way without forcing ourselves. In this way, we do not need to rely on Pseudo-*Wen* to cover up our true thoughts and actions, nor do we need approval from external things or others to prove ourselves, which is true freedom. It is worth emphasizing that "*there is no need to embellish yourself with Wen" is not that Wen isn't needed*, but to reject the formalized and superficial Pseudo-Wen that has lost touch with the Zhi. Both Laozi and Zhuangzi recognize the importance of True-Wen, which must be an expression of one's true nature and be in harmony with Zhi. Thus, they both oppose the one-sided and universal implementation of Wen. As Laozi said, "Rites are the offspring of insufficient loyalty and trustworthiness and are the beginning of chaos." (*Daodejing* Chapter38) (G. Chen 2015, p. 197) and Zhuangzi stated, "When rites and music are prevalent, the world is in chaos." Both statements oppose the universal implementation of Wen. Zhuangzi believes that what people need to do is to settle themselves. He said, "The sage behaves with integrity, yet does not deliberately display their virtue. This is a true virtue. Once artificial virtue appears, all things lose their natural essence." (*Zhuangzi Shanxing* 莊子·繕性) (G. Chen 2016, p. 413). If each person behaves properly and conceals their virtues, they will not offend others, while once they offend others, all things will lose their nature.

From the above, it can be seen that Laozi and Zhuangzi's view on the relationship between Wen and Zhi is the unity of source and function centered on human nature and emotions. In this unity, Zhi and Wen are not opposing, balanced, or complementary. On the contrary, the former is the foundation of the latter which is the manifestation of the former. The two can be understood as a unified relationship between Ti and Yong.

### 4. The Relationship of Root and Branch in Lao–Zhuang's Concepts of Wen and Zhi

Based on the Dao–Wu model, the relationship between Wen and Zhi by Laozi and Zhuangzi often involves thoughts on self-cultivation and governing the country, and is presented as the integration of root and branch. Their discussion has a strong critical consciousness towards the ritual law system, with the aim of revitalizing and optimizing the existing ritual law system.

The thoughts on self-cultivation and governing the country Laozi and Zhuangzi reflect their view of *Wen* and *Zhi*. Laozi once reminded people to value life from the perspective of self-cultivation. He said, "Which is dearer, fame or one's name? Which is more valuable, your life or your material wealth? Which is more harmful, the acquisition of fame and wealth or the loss of life? The answers are quite obvious. Therefore, the excessive pursuit of fame consumes a great deal of energy; the excessive hoarding of wealth leads to severe losses. Knowing contentment prevents humiliation, knowing when to stop avoids danger, and only in this way can one maintain longevity." (*Daodejing* Chapter 44) (G. Chen 2015, p. 225). This means that cultivating the body and mind and banishing fame and wealth is the key to self-cultivation. He teaches people to be content and to know when to stop so as to not to be lost in complex desires. This is exactly the embodiment of Laozi's emphasis on *Zhi* over *Wen*. In terms of self-cultivation and governing the country, Laozi and Zhuangzi also emphasize the importance of *Zhi*. Laozi said,

> Governing the people and serving heaven, there is nothing more important than conserving one's energy. Only by conserving energy, can one return to the *Dao* early. Returning to the *Dao* early means continuously accumulating *De*. With the constant accumulation of *De*, one is capable of both governing the people and serving heaven. If one is capable of both, it's hard to know the limits of their power. When the limits of one's power are hard to discern, one can preserve the state. Grasping the fundamental principle of preserving the state, one can ensure the state's long-lasting peace and stability. The same principle also allows one to maintain a robust physique and achieve longevity. (*Daodejing* Chapter 59) (G. Chen 2015, p. 272)

> 治人事天，莫若嗇。夫唯嗇，是謂早服；早服謂之重積德；
> 重積德則無不克；無不克則莫知其極，莫知其極；可以有國；有國之母，
> 可以長久。是謂根深固柢，長生久視之道。

"Conserving one's energy" (嗇 Sè) has the meaning of frugality, maintenance, and to cherish. As stated in *Lüshi Chunqiu Xianji* 吕氏春秋·先己, "The foundation of all actions must start with taking care of the body and cherishing one's energy." (W. Xu 2009, p. 70). Laozi believes that preserving one's virtue is the foundation of state governance. Only by laying a solid foundation and cultivating strength can we effectively respond to various challenges and make governance work smoothly. This approach also embodies Laozi's core idea that self-cultivation is the basis of state governance. Laozi also emphasizes the key points of self-cultivation in specific actions: "Steadiness controls rashness, and calmness masters restlessness. Therefore, the gentleman acts prudently all day. Even though he lives a life of wealth and honor, he can remain calm, leisurely, and detached from the mundane world. Why should the monarch of a large nation govern the world in a rash and restless manner? Rashness leads to losing oneself, and restlessness leads to losing control." (*Daodejing* Chapter 26) (G. Chen 2015, p. 158). He believes that being steady and calm are two indispensable mindsets. With such mindsets, one can remain detached and true to oneself despite wealth and glory. A ruler who can maintain tranquility and inaction in

the face of power and wealth and who is not swayed by desires can prioritize the interests of the people and the world, thus enabling the country to thrive. In Laozi's philosophy of governance, he emphasizes the importance of moral cultivation. This is why Laozi believes that Zhi should be the basis, and Wen should be secondary.

The relationship between Wen and Zhi presented by Laozi and Zhuangzi, mainly takes two forms. One is Zhi as the priority over Wen. The other is the relationship of Zhi encompassing Wen. The first form is embodied in the Dao–Wu model, which means that Dao serves as the foundation of self-cultivation and serving the country. Laozi said, "The Dao produced One; One produced Two; Two produced Three; Three produced All things," (G. Chen 2015, p. 216), which puts Dao at the position of Ben and emphasizes the priority relationship between Dao and all things in a sequential way. *Dao dejing* goes

> so, it is only after losing Dao that virtue arises, only after losing virtue that benevolence emerges, only after losing benevolence that righteousness appears, and only after losing righteousness that rites come into being. Rites are the offspring of insufficient loyalty and trust-worthiness and are the beginning of chaos. The so-called "fore-knowledge" (qianshi 前識) is nothing more than the superficial brilliance of Dao, from which ignorance begins to emerge. Therefore, a man of great virtue holds to "substance" (hou 厚) and does not dwell in "thinness" (bo 薄). He keeps his mind simple and does not reside in "superficial brilliance" (hua 華). Therefore, one should discard the "thinness" and "superficial brilliance" and adopt "simplicity" and "substance". (*Daodejing* Chapter 38) (G. Chen 2015, p. 197)

> 失道而後德，失德而後仁，失仁而後義，失義而後禮。夫禮者忠信之薄而亂之首；前識者，道之華而愚之始。是以大丈夫，處其厚不居其薄；處其實，不居其華。故去彼取此。

The sequence of *Dao* (the Way), *De* (Virtue), *Ren* (benevolence), *Yi* (righteousness), and *Li* (rites) is very clear as listed in this sentence. *Dao* is at the very beginning of the creation chain and represents the most perfect and intact state. *De*, as the offspring of *Dao*, has lost some of the information that *Dao* possesses, so the scope of *De's* information is less than that of *Dao* and cannot be used as a constraint or standard for *Dao*. By the same token, *Ren* cannot serve as a standard to constrain *De*, *Yi* cannot serve as a standard to constrain *Ren*, and *Li*, being at the end of the development chain and having lost most of the information from the previous stages, cannot serve as a standard to constrain the former three and *Yi*. According to Laozi, once *Wen*, as the core value of the social rites and laws, is separated from *Zhi*, it becomes a rigid and dogmatic regulation external to *Zhi*, thereby losing its inherent ability to reveal the *Zhi*.

Similarly, in the previous citation of Yan Zun's interpretation of *Daodejing* Chapter41, the development process of human nature can also be summarized as the sequential occurrence of "*Dao* (the Way), *De* (Virtue), *Xing* (Nature), *Qing* (Emotions), *Yu* (Desires)." Confucius said, "Without regulating harmony through rituals, it is also unworkable." But in Lao Zhuang's opinion nature is the fundamental aspect of emotion and emotion arises from "nature". Therefore, the development of emotion in the later stage cannot dictate or restrict the nature in the earlier stage. Similarly, desire cannot constrain the emotion and nature in the earlier stage of development. According to Laozi's perspective, ritual is born from sagacious foresight, a combination of the sagacious desires and ruling power, sagacious individual endeavors to universally implement this "ritual" born out of their desires, hoping to satisfy their desires for governing the world and enlightening the people. However, this approach cannot achieve the desired effect. This is because such intellect and desires are at the end of the development of one's innate disposition, and the ritual arising from them is also distant from the fundamental nature of the Dao and virtue, which means Wen cannot constrain Zhi. This is what Zhuangzi said: "Those who lose themselves in material possessions and are estranged from their true nature are called people who are

reversed." (*Zhuangzi Shangxing* 莊子·繕性) (G. Chen 2016, p. 413). Zhuangzi described the whole development process from Zhi (Ben) to Wen (Mo) by saying,

> In ancient times when perfect virtue (*zhide* 至德) prevailed, people lived together with birds and animals and mixed with everything in the world. How did they know the distinction between superior men and inferior men? All ignorant, they did not lose their virtue; all desireless(*wuyu* 無欲), they were in a state of natural simplicity as uncarved timber, which kept intact their inborn nature. When sages came into the world, persistent in their pursuit righteousness, distrust began to appear among the people. As they were indulgent in music and meticulous about rites, the world began to split. Therefore, if the natural timber is not carved, how can it be made into goblins for sacrificial rites? If the natural white jade is not broken, how can it be made into pendants? If Dao and virtue were not abandoned, why should people exercise humaneness and righteousness? If men's inborn nature were not discarded, why should people resort to rites and music? If the five basic colors (*wuse* 五色) aren't confused, why should people need colorful designs? If the five basic notes are not confused, why should people need the six harmonies (*liulv* 六律)? To carve the timber into vessels is the fault of an artisan, and to destroy Dao and virtue for the sake of humaneness and righteousness is the error of a sage. (*Zhuangzi Mati* 莊子·馬蹄) (G. Chen 2016, p. 258)

> 夫至德之世，同與禽獸居，族與萬物並，惡乎知君子小人哉！同乎無知，
> 其德不離；同乎無欲，是謂素樸。素樸而民性得矣。及至聖人，蹩躠為仁，
> 踶跂為義，而天下始疑矣；澶漫為樂，摘僻為禮，而天下始分矣；故純樸不殘，
> 孰為犧尊！白玉不毀，孰為珪璋！道德不廢，安取仁義！性情不離，安用禮樂！
> 五色不亂，孰為文采！五聲不亂，孰應六律！

From the above, we can see that Laozi and Zhuangzi believe that Ren, Yi, Li, Yue (Music), etc., which lose Dao and De are no longer the original ones inherent in human nature. If they use their externalized form, Pseudo-Wen, as the criterion for self-cultivation and governing the world, then they are even further away from Zhi. As Wang Bi said, "Renyi originates from within. It is still false to act on it, let alone focusing on external decorations for a long time." Pseudo-Wen is the external form of the Wen for the continuous loss of the Dao. It is also the result of deviating from nature and following desires as Zhuangzi said, which represents the extreme end. Therefore, seeking Wen to examine Zhi is to search beyond one's inherent nature, which can never achieve the goal of returning to one's true nature. Therefore, if you want to seek Zhi, you must get rid of the bondage of Pseudo-Wen, dwelling in the essence, not lingering in the superficial, returning to the origin from the end, and reducing selfishness and desires.

Secondly, Zhi encompassing Wen also means that One incorporates Many. This is reflected in the critics of Laozi and Zhuangzi to rites and the law system. Laozi said "Rites are the offspring of insufficient loyalty and trustworthiness and are the beginning of chaos. The so-called foreknowledge is nothing more than the superficial brilliance of Dao, from which ignorance begins to emerge. Therefore, a man of great virtue holds to substance and does not dwell in superficiality and insubstantiality. He keeps his mind simple and does not reside in superficial brilliance. Therefore, one should discard the superficiality and insubstantiality and adopt simplicity and substance." In this statement, the dogmatic system of rites and music is considered a fore-knowledge, which is the behavioral norm that has been pre-determined before people have real feelings and actions in their situation. This norm lacks people's real emotional experience, so Laozi calls it the false doctrine. He criticizes these pre-set rites as Pseudo-Wen. We can infer that he agrees with the non-pre-set rites, that is, the words and deeds that people naturally express in real situations along with their real emotional experience, which is True-Wen. This kind of Wen is under the control of Zhi. However, it is worth noting that since these real rites are based on the individual, specific, and real emotional experience, then rites should be diverse rather than a standardized pattern. Taking mourning rites as an example, Laozi says, "After win-



ning a battle, one should handle it with the ceremonial rites of mourning." (*Daodejing*, Chapter 31) (G. Chen 2015, p. 177). He agrees with the education of the people with Wen. However, there are regulations, like "In ancient times, funerals had rituals based on social status, and ranks were observed. Specifically, the emperor's coffin was seven layers, the princes' were five layers, the senior officials' were three layers, and the lower-ranking officials' were two layers." (*Zhuangzi Tianxia* 莊子·天下) (G. Chen 2016, p. 861). This form of the mourning rites is regulated by various normative systems, which restrict people's actions and words. Under such regulation, people's real feelings are obscured and damaged, and they gradually lose their nature. Therefore, there is doubt and confusion in *Zhuangzi Tianxia* 莊子·天下 (G. Chen 2016, p. 201), where Zigong 子貢 asked Mengzifan 孟子反 and Ziqinzhang 子琴張 whether it was appropriate to sing in front of the corpse. The two men replied to him saying he "does not know the meaning of the rites." Here, the "meaning of the rites" is the inherent quality of rites which should be a certain way of speech and action naturally produced by people's emotional experience for the deceased. If it is bound by the form of *rites* at the end, it loses its educational function for human nature. Laozi and Zhuangzi praise simplicity while criticizing externalization, standardization, ceremony, or any other types of ornamentation (D'Ambrosio 2020, p. 7). Simplicity refers to the experience of human nature that integrates with virtues. Words and deeds based on this experience are the reflection of *rites* and belong to True-Wen. In addition, the True-Wen revealed by individual nature can vary from person to person and from situation to situation. For example, in terms of the passing away of relatives and friends, there are different ways of expression. Zhuangzi mourned his deceased wife, singing while beating a pot (*Zhuangzi Zhile* 莊子·至樂) (G. Chen 2016, p. 457). when Zhuangzi died, Qin Shi 秦失 ran to his coffin, let out three long wails towards the sky and then left without shedding a tear. Zhuangzi laments for Hui Shi's 惠施 death, saying, "Ever since Hui Shi departed from this world, I have no worthy adversary left! I have no one to debate with anymore." (*Zhuangzi Xuwugui* 莊子·徐無鬼) (G. Chen 2016, p. 646). It can be seen that the Wen that matches Zhi is diverse and specific, showing a One-to-Many 一對多 relationship of Zhi encompassing Wen. The reason why Laozi and Zhuangzi criticize the rigid rites and law system is to awaken its flexibility which should be adjusted at any time according to time, place, and people, thus breaking the single and dogmatic nature of the rigid rites and law system and injecting vitality into it.

To sum up, in the relationship between Wen and Zhi, the latter occupies the absolute dominant position. Some scholars believe that Laozi and Zhuangzi favor Zhi over Wen, which means that they aim to abandon or eradicate all forms and place them at opposing ends. However, this is not the truth. What Laozi and Zhuangzi seek to remove is Pseudo-Wen, while for True-Wen, they advocate incorporating it within Zhi. Whether Laozi's expressions of banishing wisdom, sophistry, deception, cunning, and profit, or Zhuangzi's expression of rhetoric annihilating nature and knowledge drowning the soul, these all emphasize the subjectivity and root position of Zhi in order to highlight the dominance and priority of Zhi. Laozi and Zhuangzi do not absolutely put Wen and Zhi at the two opposing ends, but view them as a unified whole. In their view, without first clarifying the dominant position of Zhi, Wen has no basis or necessity to exist, because Wen arises from Zhi, and Zhi must firstly truly exist for Wen to acquire meaning. As Wang Bi said, "truth originates from the essence of things; the foundation comes from the simplicity of things." (B. Wang 1980, pp. 191–92). What Laozi and Zhuangzi seek to abandon or eradicate is the Wen that has detached from Zhi and become independent of Zhi. This is the core viewpoint of Laozi's holistic perspective on essence and perception.

On this premise, Laozi and Zhuangzi are not opposed to humanistic education. Laozi only objects to the binding and constraining of people by ritual music systems that have become externalized and disconnected from human nature. They believe that such systems are insufficient to correct human nature. Zhuangzi, like Laozi, also believes that feelings that naturally occur are in accord with nature. These feelings do not mix with selfish desires and arise naturally in response to external things. Zhuangzi believes such genuine

feelings can be allowed to develop freely, as stated in *Zhuangzi Pianmu* 莊子·駢拇, "What I refer to as 'good' is not merely about benevolence and righteousness; it simply implies embracing the natural inclinations and fate of one's being." Therefore, any form of expression that flows from genuine feelings and emotions is considered True-Wen. Through advocating for the harmonious True-Wen that aligns with the Zhi, and endorsing the diversity of its manifestations, Laozi and Zhuangzi highlight their awareness of critiquing the outdated and rigid social norms that externalize human nature. Through this, they also express their belief in the fundamental importance of self-cultivation as the essence of governance. In the view of Laozi and Zhuangzi, social norms and systems employed for governance should not detach from or contradict the fundamental needs of the people. Correct and rational social norms and systems should be adjusted and updated based on the differences among individuals, circumstances, and locations, in order to be in line with the practical situation. Otherwise, these social norms and systems will become authoritarian, rigid, and dogmatic, alienating the people and eventually leading to confusion in people's hearts and chaos in society.

## 5. Conclusions

In the research field where the Confucian view of Wen and Zhi is mainstream, not only has the unique characteristics of the Daoist view not been fully noticed, but it has also suffered from numerous misunderstandings. The connotations of Wen and Zhi as defined by Confucius through Li (rites) and Ren and Yi (benevolence and righteousness) are fundamentally different from those of Laozi. If applied directly without clarification, it will not yield the desired research results. To avoid misunderstandings, this article first clarifies the concepts of Wen and Zhi in Laozi and Zhuangzi's philosophy. Based on the Dao (the way)–Wu (object) model in their thoughts, Zhi represents the natural state of human's innate simplicity and desirelessness, deriving from Dao (the Way) and De (virtue). Wen, as the specific deployment and presentation of Dao, De, and Xing (nature), refers to the corresponding expressions of speech and behavior arising from Zhi. Secondly, upon establishing the inherent meanings of Wen and Zhi, we further analyse the ideas related to Xing and Qing (emotions), thereby clarifying the close relationship between the view of Wen and Zhi in Laozi and Zhuangzi's thoughts and human's nature and emotions. As a specific embodiment of the Wen–Zhi relationship, the relationship of Xing and Qing further illustrates the relationship of source and funciton as the unity. Finally, when Laozi and Zhuangzi apply the relationship of Xing and Qing to self-cultivation and governing the state, the corresponding relationship between Wen and Zhi presents as a unity of Ben (root) and Mo (branch), where Zhi is the root, and Wen is the branch. Once we clarify these characteristics, we have a new understanding of the Zhi-centric feature of Laozi and Zhuangzi, which is that their Zhi is in a dominant position, and Wen is unified under Zhi in this premise. This is not a rejection of Wen, but rather, it integrates Wen with Zhi. This relationship dissolves the Pseudo-Wen that has deviated from Zhi, while at the same time, it allows the True-Wen naturally revealed from Zhi to play its real value and role.

The characteristics of Laozi and Zhuangzi's view on Wen and Zhi differ from Confucius' view, which embodies balance, harmony, and mutual restraint. They have a pronounced critical consciousness. However, this critical consciousness is not to completely overthrow Wen but to discard the false and retain the true to revitalize and update it. In Laozi and Zhuangzi's philosophy, there are many criticisms of ceremonial etiquette. If we apply Confucius' view on the concept of ceremony to the interpretation of Laozi and Zhuangzi's view on Wen and Zhi, it is easy to see it as an ideological tendency that completely opposes ceremonial etiquette, social ethical norms, and pursues absolute freedom. However, Laozi and Zhuangzi's source and funciton as the unity and root and branch as the integration fundamentally hope to reflect and update the increasingly rigid ceremonial forms that are out of date. This implies maintaining a vigilant attitude at all times, scrutinizing cultural forms that affect human nature. The purpose of this is to keep the essential cultural, educational, and managerial methods beneficial to people's life needs vital in the

long term, and to help people explore the ideal way of existence that suits their development. From this perspective, the views on Wen and Zhi in Confucianism and Daoism are not conflicting, and both aim at better development for people. The critical spirit of Daoism's view on Wen and Zhi, as well as its characteristic of Zhi governing Wen and Wen revealing Zhi, advocates for the flexibility and diversity of Wen under the control of Zhi. This had a profound influence on the great development of literary forms in the discourse of them during the Wei, Jin, and Northern and Southern Dynasties period (220–589 AD)..

Prior to the Wei and Jin dynasties, emphasis was often placed on the content in literary expressions, while the rhetorical forms were relatively simple. For example, Liu Xiang 劉向 (about 77 BC–6 AD) in the Han Dynasty (202 BC-220 AD) stated, "Pursuing Zhi as the dominant and Wen as the secondary makes a saint" ([Liu 1987](), p. 516). This understanding is quite in line with the Daoist perspective on Wen and Zhi. Dong Zhongshu 董仲舒 (179–104 BC) said, "When both Zhi and Wen are equipped, the rites are complete." ([Su 1992](), p. 27). This highlights the characteristics of the Confucian view of Wen and Zhi during the Han Dynasty. By the Wei, Jin, and Northern and Southern Dynasties period, the value of Laozi and Zhuangzi's thoughts was reassessed, and their approach of letting Zhi govern Wen gradually gained attention. Wang Bi's "sages have emotions" (*Shengren youqing* 聖人有情) (B. [Wang 1980](), pp. 191–92) theory and his advocacy for "nature and emotions are interdependent" (B. [Wang 1980](), p. 217) gradually helped people understand the form of Wen. Influenced by Wang Bi's thoughts, people believed that as long as it was in line with nature, external emotions were under the natural domination of nature. This greatly broke the limitations and constraints of the simple form of Wen. Liu Xie 劉勰 (about 465–532AD) later proposed "The thoughts should be rich and the language should be cultured, the feelings should be sincere and the words should be clever and exquisite. These are the basic rules of writing." ([Fan 1978](), p. 15). With Liu Xie's *The Mind of Literature and Carving Dragons* (*Wenxin diaolong* 文心雕龍) as a landmark, the style of literature and the literary trends become greatly enriched thereafter. Chen Liangyun 陳良運 believes that the Wei, Jin, and the Southern and Northern dynasties period was a time when Chinese pure literature entered a completely self-conscious era, laying a theoretical foundation for the golden age of ancient Chinese literature in the Tang dynasty and beyond. The concept of Wen and Zhi in Daoism has a rich theoretical connotation and value (L. [Chen 2001](), pp. 111–12). Its characteristic of criticizing Pseudo-Wen and advocating True-Wen can be applied to many research fields, such as culture, society, politics, ethics, etc., and inject vitality into these studies.

**Author Contributions:** Conceptualization S.J.; writing—original draft preparation, S.J.; writing—review and editing, Y.H.; funding acquisition, Y.H. All authors have read and agreed to the published version of the manuscript.

**Funding:** This research was funded by National Social Science Fund of China, grant number: 21AZJ 005.

**Conflicts of Interest:** The authors declare no conflict of interest.

## Notes

[1]　Currently, the term "literary embellishment" is commonly used to refer to "文" and "basic stuff" is used to refer to "質." (see ([Guo 2002](), Zhongguozhexue da cidian 漢英中國哲學大辭典, p. 179). These terms were originally used by literati in the Wei and Jin 魏晉 dynasties, who used the fixed term "文質" to evaluate poetry and literature. Modern scholars, influenced by Western literary concepts, have equated this term with the form and content of a text, further narrowing the scope of the concept of "文質." However, in early Daoist thought represented by Laozi and Zhuangzi, the concept of "文質" is rich and complex. Therefore, using "literary embellishment" and "basic stuff" to refer to "文" and "質," respectively, may lead to confusion. Hence, in this article, we use the Chinese pinyin of Wen and Zhi to denote these two concepts, aiming to provide a clear and understandable explanation for readers.

[2]　*The Book of Poetry Tianbao* 詩經·天保 states, "the people are simple and honest, satisfied with a full meal each day. Whether officials or civilians, all are touched by your grace." ([Ruan 2009](), p. 881) This reflects the importance of the king's virtue in awakening the nature of the people. *The Book of Documents Yugong* 尚書·禹貢 mentions, "within three hundred miles, governance is carried

out through cultural education", (Ruan 2009, p. 322) indicating that cultural education was considered an important national policy. *The Book of Changes Bigua* 周易·賁卦 says, "observing celestial phenomena allows us to understand the changes in seasons; observing human culture enables us to educate and transform the world." (Ruan 2009, p. 75)

3    During Jiang Xichang's 蔣錫昌 annotating of the *Daodejing*, he used the phrase polished by Wen from the *Discourse of the States Zhouyu* (國語·周語), combined with Wei Zhao's 韋昭 (204–273 AD) commentary on this chapter which states "Wen refers to ceremonial rituals." (S. Li 2014, p. 663)

4    According to Yan Buke (B. Yan 2015, pp. 284, 291), the interactive and balanced relationship between Wen and Zhi is due to the regulatory function of rites (li 禮); Li originates from human nature and human emotions, and also regulates them. When Zhi is dominant, it is rectified by Wen; when Wen is dominant, it is rectified by Zhi. He also believed that in the political realm, the interplay between Wen and Zhi constitutes a necessary tension that balances and harmonizes the pursuit of Confucian ideology with the reality of despotic bureaucratic politics.

5    According to Lu Shuangzu 陸雙祖 (Lu 2016, p. 17), it was through Confucius' 孔子 (551–479 BC) emphasis on this relationship and his core idea of "wen and zhi are harmoniously blended (wenzhi binbin 文質彬彬)," developed and refined further by Xunzi 荀子 (313–238 BC), that the orthodox position of the Confucian view on this matter was established.

6    Chen Qingkun 陳慶坤 (Q. Chen 1995, p. 31) believed that Confucius's view on Wen and Zhi embodied an exemplary personality characterized by the middle way (zhongdao 中道) between them. When summarizing the characteristics of ancient Chinese views on Wen and Zhi, Hua Jun 華軍 (Hua 2020, pp. 53–59) pointed out that they can be generalized as a concept based on the harmonious development of both natural and cultural elements, rooted in the mutual complementation of Wen and Zhi (wenzhi huyong 文質互用) as a basic practice model, and as a notion of pursuing the middle path between Wen and Zhi (wenzhi zhongdao 文質中道).

7    This article discusses the concept of Laozi and Zhuangzi based on the ideas presented in the texts *Daodejing* and *Zhuangzi*. However, it does not delve into the relationship between Laozi and Zhuangzi and the texts *Daodejing* and *Zhuangzi*.

8    During the Song Dynasty (960–1279 AD), Fan Yingyuan 范應元 (lived approximately between the years 1241 and 1252) was influenced by this above interpretation and definitively explained this line as "Wen is inadequate to transform the people, so one should return to their roots; Wen should not overcome Zhi." He pointed out the relationship between Wen and Zhi expressed by Laozi, believing that Wen refers to the norms of education, and Wen should not surpass Zhi (S. Li 2014, p. 664). Lü Huiqing 呂惠卿 (1032–1111 AD) of the Ming Dynasty also believed that sagacious wisdom, benevolence and righteousness, and cunning and profit all belongd to Wen rather than Zhi. He noted that Laozi opposed Wen and Zhi, which Wen is insufficient to bring people to an ideal state, hence this view of Wen should be completely abandoned (S. Li 2014, p. 666).

9    Gao Heng 高亨 (S. Li 2014, p. 633) believed that the pronunciations of the ancient characters wei (為) and wei (偽) were the same, while Yu Xingwu 于省吾 (S. Li 2014, p. 633) believed that wei (為) and wei (偽) were interchangeable characters; their view leads to a conclusion that favoring wen (為文) should be interpreted as "Pseudo-wen" (weiwen 偽文) (S. Li 2014, p. 633).

10    Guayu (寡欲) refers to the degree of reducing personal desires and returning to the original state of nature and emotions. In the *Zhuangzi Gengsangchu* 莊子·庚桑楚, it also mentions the state of desires reversing your nature and emotions, which is equivalent to the meaning of Guayu. This sentence shows us Zhuangzi's understanding of the sequence of the emergence of personal desires, that is, from nature to emotions, and then from emotions to desires. Therefore, the path to return to simplicity should be from desires to emotions, and then to nature.

11    Wang Bi's full commentary to the Daodejing chapter19 reads, "Sagehood (*sheng* 圣) and wisdom (*zhi* 智) designate the best (*shan* 善) of human talent; benevolence and righteousness designate the best of human behavior and cleverness and sharpness designate the best of human resources. However, the text directly says that these should be repudiated. Because such "decoration" (*wen* 文) is utterly inadequate, one does not give people the chance to identify with these expressions and so never does anything that exemplifies what they mean. Thus the text says: Because these three pairs of terms serve as mere decoration, they are never adequate. When allowing people to identify with something, let them identify with your simplicity and minimal desires." 聖智，才之善也；仁義，人之善也；巧利，用之善也。而直云絕。文甚不足，不令之有所屬，無以見其指。故曰，此三者以為文而未足，故令人有所屬，屬之於素樸寡欲。(Lynn 1999, p. 82)

12    Li Shuihai 李水海 believes that the character "抱" (*bào*) had a dialectal pronunciation of "耦" (*ou*) in the State of Chu during the Warring States period, which meant unification or combination. "素" refers to undyed silk cloth according to the Shuowen jiezi 說文解字, while "樸" means unprocessed wood (S. Li 2014, p. 666).

13    Ye Shuxun 葉樹勳 believes that early Daoists regarded humans as one of the many beings in the universe. When viewed from a broader perspective, human problems and those of other beings share similar roots. Therefore, they expanded their approach to solving human problems to encompass the entire universe, constantly tracing phenomena and hoping to find the root cause of the creation and essence of all things. They hoped to find a fundamental plan for guiding human behavior at the root of the universe. As such, they continuously traced back to explore the origin of the universe, in order to provide basic guidance for solving human problems (Ye 2017, p. 31). Unlike the way, Sinologists usually use "correlative cosmology" to summarize the unique cosmology of ancient China (Matthews 2021, pp. 260–71), the Dao–Wu model of cosmology emphasizes the correlation between nature and human affairs, rather than being reflected through one-to-one correspondences of categories.

14 According to Chen Zhijun's 陈志軍 point of view, emotion is the direct manifestation of nature and the concrete display of nature. In fact, through the experience and expression of emotion, we can cultivate and shape nature. Therefore, it can be said that the nurture and cultivation of nature are actually reflected in emotion (Z. Chen 2012, p. 136).

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
