# Peer review of "A New Study on Features Exploring of the Concept of Wen and Zhi in Lao-Zhuang’s Philosophy"

_religions, doi:10.3390/rel14081013_

Round 1

Reviewer 1 Report

A fascinating topic, particularly in term of comparative philosophy. Readings who lack specialization in Chinese language will definitely benefit from the analysis and resources cited.

Some references are missing in the text, including references for direct quotes.

Some quotations are repeated multiple times.

There is a tendency to overemphasize the evidence for claims being made, using such as "clearly." Include sufficient evidence to support that level of certainty or use more moderate phrasing.

Review overall syntax and clarity of text.

Please provide a more in depth explanation of "Dao-things" and additional textual evidence for the "root and branch" analogy.

Author Response

Suggestion 1. Some references are missing in the text, including references for direct quotes.Some quotations are repeated multiple times.

  1. Additional primary and secondary sources have been added to provide support for the arguments presented.
  2. Avoid repeating quoted content as much as possible, except for essential information.

Suggestion 2.There is a tendency to overemphasize the evidence for claims being made, using such as "clearly." Include sufficient evidence to support that level of certainty or use more moderate phrasing.

  1. The logical flow of the entire text has been reorganized, and additional evidence has been provided. Please translate this sentence into English.
  2. Strive to ensure sufficient evidence, for example by incorporating direct quotations from Laozi's original text to establish a chain of evidence, thereby avoiding the excessive emphasis on words like "clearly."

Suggestion 3.Review overall syntax and clarity of text. Please provide a more in depth explanation of "Dao-things" and additional textual evidence for the "root and branch" analogy.

(1) From line 400 to 428 of the article, a more detailed explanation is provided for the "Dao-Object" mode.

(2) From line 579 to 627 of the article, additional evidence is presented for the correlation between self-cultivation and governing the country in terms of the "origin-end" relationship.

(3) From line 665 to 719 of the article, further evidence is added regarding the progressive development of "Dao," "De," "Xing," "Qing," and "Yu" in relation to the "origin-end" relationship.

Reviewer 2 Report

It is very interesting to explore the concept of wen and zhi in lao-zhuang philosophy, especially when most relevant discussions are centered around Confucianism. The author goes beyond the Confucian framework to discuss the relevant issues, which is very rare.

However, there are a lot of typos, technique problems and issues about the language in this article, which needs to be corrected. Following are just some of them:

Typos:

1.Line 53, “in the Aristotle” should be “in the Analects”.

2.Line 177 “刻画的纹饰” should add pinyin.

3. Line 250, “he saide” should be “he said”.

4. Line 306, the pinyin is “yingdiwang” while the Chinese characters of it is “庚桑楚”.

Technique Problems:

1.Usually, the citations of the Analects are marked by numbers, instead of the name of the chapters.

For examples, the citation of line 53 can be referred as “the Analects 6.18”. 

2. The translation of the title of the same book should be constant. Line311, 316,386, 411 translated 《庄子》as “the book of master Zhuang” while other places translated it as “the zhuangzi”.

3. The author should indicate the source of the translations of the classical texts. 

Furthermore, the author may want to reconsider some of the expressions, for example:

1. Line 61-62,” natural nature”, “benevolence and righteousness” virtue.

2. Line 164-166, On the surface, it seems both Confucius and Lao-Zhuang are discussing the perspec- 164 tives of Confucius and Lao-Zhuang's “wen” and “zhi” are discussing the relationship be- 165 tween a person's inner essence and external form. 

Author Response

1.Suggestion . However, there are a lot of typos, technique problems and issues about the language in this article, which needs to be corrected. Following are just some of them:

Response: I have consulted a professional translator who has reviewed and edited the entire text to minimize errors that may occur during text editing.

2.Typos:

  • Line 53, “in the Aristotle” should be “in the Analects”.

Response:The paper has been revised.

  • Line 177 “刻画的纹饰” should add pinyin.

Response:The paper has been revised.

  • Line 250, “he saide” should be “he said”.

Response:The paper has been revised.

(4)Line 306, the pinyin is “yingdiwang” while the Chinese characters of it is “庚桑楚”.

 Response: The paper has been revised.

3.Technique Problems:

  • Usually, the citations of the Analects are marked by numbers, instead of the name of the chapters.

Response: In the paper, all citations of "The Analects" were converted to utilize numerical annotations.

  • The translation of the title of the same book should be constant. Line311, 316,386, 411 translated 《庄子》as “the book of master Zhuang” while other places translated it as “the zhuangzi”.

Response: The consistent citation format has been applied to the same book.

4.The author should indicate the source of the translations of the classical texts.

Response: Annotations indicating the textual sources of all the classical references cited in the paper have been added.

5.Comments on the Quality of English Language Furthermore, the author may want to reconsider some of the expressions.

Response: In accordance with reference sources, concepts and terms mentioned in the paper have been cross-checked and revised. Improvements have also been made in the presentation of arguments, aiming to achieve smooth writing and clear expression of ideas.

Reviewer 3 Report

The main argument itself, that wen and zhi are not opposed to each other is an interesting one. However, the structure of argumentation is somewhat incoherent and the bibliography is very limited. I suggest the author work on restructuring the paper, starting with the abstract, clarify what they are gonna fo and why it is important. Moreover, I suggest the author look to Wei-Jin philosophy as this was the primary goal during this time. 

Incoherent

Author Response

Suggestion: I suggest the author work on restructuring the paper, starting with the abstract, clarify what they are gonna fo and why it is important. Moreover, I suggest the author look to Wei-Jin philosophy as this was the primary goal during this time. 

Response:

Thank you very much for your valuable suggestions. I have made the revisions to the paper according to the suggestions provided. I kindly request you to review the revised version.

After revision, the paper has been organized as follows:

Research background: Currently, scholars' research on the "Wen(inner essence文)and Zhi(external form质)" in the history of Chinese thought focuses mainly on the Confucian perspective, with less attention given to the Daoist perspective. Even when the Daoist perspective is occasionally mentioned, it is often a mere application of the understanding of literary and moral qualities from Confucianism to Daoism. Thus, there is a need for scholarly writing to clarify the rich and unique aspects of the Daoist perspective on literary and moral qualities, which often get undervalued.

Chapter 1: Using "Laozi" and "Zhuangzi" as a reference, this chapter delves into the definition of "literary" and "moral" qualities.

Chapter 2: From the perspective of the relationship between form and function, this chapter explores the connection between Daoist views on literary and moral qualities and human nature.

Chapter 3: From the perspective of the relationship between means and ends, this chapter examines the relationship between Daoist views on literary and moral qualities and self-cultivation and governance.

Conclusion: From the perspective of the influence of Lao-Zhuang thought on later generations, this section briefly discusses the impact of Lao-Zhuang's literary and moral qualities on the significant development of the Wei-Jin literary and moral discourse.

Round 2

Reviewer 3 Report

Manuscript has been improved and can be accepted as is